# Development and application of decontamination methods for the re-use of laboratory grade plastic pipette tips

Sang Hyuk Lee[1,2,3], William Kastor[2,3], Xiao Fu[2,3], Vikas Soni[4], Michael Keidar[4], Marc Donohue[1], Steve Wood[2], Enusha Karunasena[2]*

1 Department of Chemical and Biomolecular Engineering, Whiting School of Engineering, Johns Hopkins University, Baltimore, MD, United States of America, 2 Division of Biology, Chemistry, and Materials Science, Office of Science and Engineering Laboratories, Center for Devices and Radiological Health, US Food and Drug Administration (FDA), Silver Spring, MD, United States of America, 3 Oak Ridge Institute for Science and Education, Office of Science, US Department of Energy, Oak Ridge, TN, United States of America, 4 Department of Mechanical and Aerospace Engineering, School of Engineering and Applied Science, George Washington University, Washington, DC, United States of America

* enusha.karunasena@fda.hhs.gov, enusha.karunasena@gmail.com

**Data Availability Statement:** All relevant data are within the manuscript and its Supporting Information files.

## Abstract

During the SARS-CoV-2 pandemic, a need for methods to decontaminate and reuse personal protective equipment (PPE) and medical plastics became a priority. In this investigation we aimed to develop a contamination evaluation protocol for laboratory pipette tips, after decontamination. Decontamination methods tested in this study included cleaning with a common laboratory detergent (2.5% Alconox® solution followed with steam decontamination), exposure of ozone vapor at 250 and 14400 PPM * minute, and exposure to cold atmospheric plasma (CAP). All tips (control and experimental groups) were introduced to the methods described, while tips exposed to DNA extracts of *Aeromonas hydrophila* (ATCC-23211) were assessed for experimental groups. Decontamination was determined by turnover ratio and log reduction in detectable genomic material on the contaminated products using real-time quantitative PCR (qPCR) assay. Our results showed, cleaning tips with lab detergents along with steam decontamination removed genetic material, resulting in the highest log reduction, compared with ozone or CAP treatments. Detergent/washing methods showed the second highest turnover ratio (95.9%) and log reduction (5.943). However, the excessive residue (post-cleaning) on the plastic, within inner filters, and tip boxes suggested that washing with lab detergents was not favorable for reuse. Ozone vapor at 14400 PPM * minute showed the highest turnover ratio (98.4%) and log reduction (4.511). CAP exposure with tips inverted (the tip end exposed closer to the plasma flame) for 1 minute showed a turnover ratio of (68.3%) and log reduction (4.002). Relatively, lower turnover ratio and log reduction of CAP could be improved by optimization, such as increasing the exposure time. Future testing would provide fine-tuned conditions for CAP-specific decontamination of plasticware. In this study we were able to provide fundamental insight into a non-traditional decontamination method for single-use plasticware that could render these products reusable.

**Funding:** We would like to thank the ORISE fellowship program for supporting scholars who contributed to this study and FDA Office of the Chief Scientists (OCS)/Medical Counter Measures (MCMi)/OCET and FDA/OST for their funding and support of this study.

**Competing interests:** The authors have declared that no competing interests exist.

## Introduction

The recent pandemic exhausted medical plastic supply-chains. For these reasons, decontamination methods were proposed to mitigate PPE shortage, tested methods included ultraviolet light [1–3], methylene blue [4], and hydrogen peroxide [3, 5, 6] vapor. In these experiments we compared the efficacies of streamlining methods, including washing pipette tips with laboratory detergents followed with steam decontamination, exposure to ozone vapor and cold atmospheric plasma. The aims of this study were to identify nondestructive, efficient, and effective decontamination methods for removing genetic material from pipette tips.

### Decontamination modality one

Laboratory cleaning detergents are commonly used for glassware and plastics. We tested the use of Alconox®, which contains sodium tripolyphosphate as water softener, sodium alkyl-benzene sulfonate as a foaming agent, and tetrasodium pyrophosphate as stain remover. A study by Luijt et al. in 2001, tested a comparable reagent (liquinox) in combination with CIDEX® decontamination to evaluate the clearance of two RNA viruses from twenty ethylene oxide sterilized 5F balloon catheters that were inoculated and simulated through 'use' and 're-use'. Following qRT-PCR analysis and culture assays neither liquinox nor CIDEX were able to fully re-clean catheters [7] and remove the viruses. In contrast, an investigation conducted by Shields et al. used various cleaning solutions to evaluate the recovery of *Cryptosporidium parvum* and *Cyclospora cayentanensis* oocytes from contaminated food items. This investigation demonstrated increased oocyte recovery (97.2%) when using a 1.0% Alconox® solution [8]; which is a widely used laboratory cleaner, with 1% peroxide-based bleach.

### Decontamination modality two

Ozone is triple oxygen molecule and act as a powerful oxidant. Based on the CDCs August 2003 "Guidelines for disinfection and sterilization in healthcare facilities", exposing atmospheric $O_2$ to ultraviolet (UV) radiation to activate ozone is approved in decontamination for clinical settings, and by the US FDA. Yet, industrial ozone decontamination is still an evolving technology with restrictive requirements [7], such as high humidity. However, conditioned ozone decontamination inactivated airborne respiratory viruses [8], on various [9] surfaces. In this study we evaluated the clearance of residual DNA on pipette tips as a measure of ozone decontamination.

### Decontamination modality three

CAP is a thermal non-equilibrium state between heavy positive ions and electrons achieved [10, 11] through rapid atmospheric pressure discharge. A previous study characterized CAP devices for reactive species [12] using optical emission spectroscopy. CAP maintains temperatures of 25–45˚C [11, 12] and generate reactive oxygen and nitrogen species (ROS/RNS) which exhibit both [10, 13] virucidal and bactericidal properties. This study evaluated the efficacy of CAP decontamination and clearance of residual DNA on plastic pipette tips.

This investigation compared the efficacies of genetic material removal from pipette tips commonly used for molecular assays, like polymerase chain reaction (PCR). The decontamination methods tested are a common laboratory detergent and steam sterilization, ozone vapor, and CAP. Commonly used laboratory and industrial applications were replicated [11, 12, 14, 15] to compare decontamination modalities.

The biological 'contaminant' for this study were DNA extracts from *Aeromonas hydrophila* (ATCC-23211). *Aeromonas hydrophila* is a gram-negative, opportunistic, facultative, anaerobic

bacilli that is widely used as a model organism due to its route of fecal-oral [16] transmission. *Aeromonas sp*. infections have a broad range of clinical presentation including bacteremia, hepatobiliary infection, pneumoniae, and skin /soft tissue infections, but most commonly manifest in acute [16, 17] gastroenteritis. In this study, *A. hydrophila* was used as the model organism due to its ubiquitous and pathological properties.

## Materials and methods

This investigation (Fig 1) used Biotix® Utip Filtered packed 10 μL pipette tips (Biotix Inc, CA) as a representative pipette tip. The volume of fluid tested was 8.0 μL and for experimental conditions, this included extracted DNA from *Aeromonas hydrophila* (ATCC-23211). After exposure to genomic materials, the tips were re-racked in the pipette tip box and incubated for 15–20 minutes at room temperature in a BSL-2 biosafety cabinet. Following this incubation period, tips were exposed to a standardized decontamination protocol, relative to the modality tested. Post-decontamination, tips were reused to harvest residual genetic material dispensed into 8μL of sterile ultrapure DNase/RNase free water, nuclease free water was aspirated thrice from each tip to maximize harvesting, (Thermo-Fisher Scientific, MA)—samples were then tested for residual genomic material using qRT—PCR.

qRT-PCR was used to pinpoint an acceptable range for threshold cycle ($T_c$) for sterile (negative control) and DNA-contaminated (positive control) groups. Experimental groups were treated with decontamination methods and evaluated for DNA, results were compared with Tc from negative and positive controls. This extrapolation was useful for comparisons between

## *Protocol Outline*

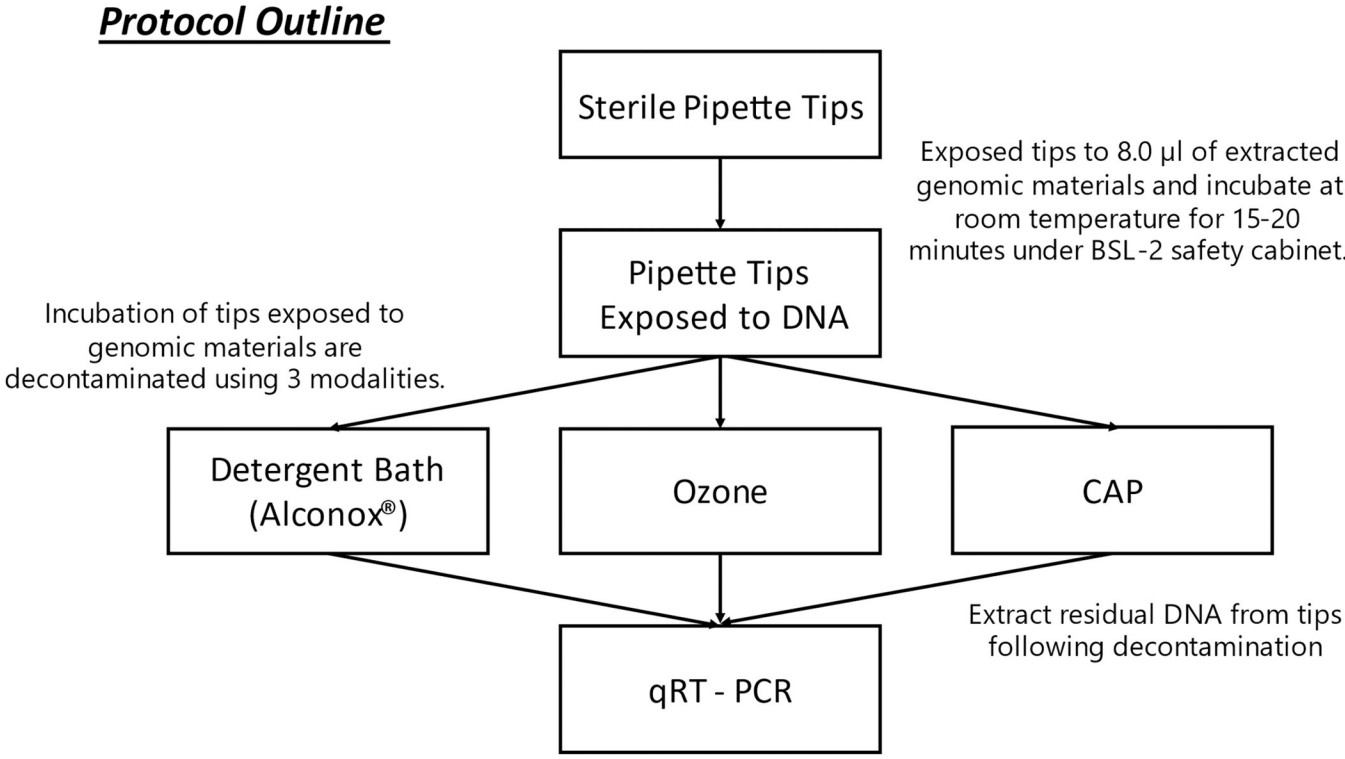

**Fig 1. Protocol outline.** Fig 1 showed the overall experimental design of the investigation using three different decontamination modalities: detergent bath, ozone, and CAP. Experimental samples were prepared by exposing sterile pipette tips to 8.0 μL of extracted genomic materials and incubated at room temperature for 15–20 minutes under BSL-2 safety cabinet. Exposed pipette tips were decontaminated either via detergent bath, ozone, or CAP. Residual genomic material were collected from the decontaminated tips for quantification via qRT-PCR.

bacterial plate counts along with qRT–PCR $T_c$ results, as $T_c$ showed the number of doubling cycles needed to amplify DNA copies to detectable range.

## Microbial culture

Inoculated cultures of *A. hydrophila* (ATCC-23211) were suspended in 0.5 mL of sterile nutrient broth (NB; Thermo Fisher Scientific, MA). The suspension was aseptically transferred to a 15 mL conical test tube containing 4.5 mL of sterile nutrient broth and allowed to incubate overnight at 37˚C with shaking at 150 rotations per minute (rpm) (Southwest Science, NJ).

## Bacterial growth calibration curve

A subculture was prepared in 100 mL of sterile NB in a 250 mL Erlenmeyer flask. One hundred microliters of overnight culture were inoculated into the subculture flask and incubated at 37˚C with shaking at 150 rpm. Growth was measured spectrophotometrically, a cuvette with 1.0 mL of sterile NB was used as a blank for measuring optical density at 600 nm ($OD_{600}$) using the GENESYS™ 10S UV-Vis Spectrophotometer (Thermo Fisher Scientific, MA). Samples of the subculture were measured at 0, 1, 3, 5, 6, 7, 8, and 9 hours respectively, by aliquoting and measuring $OD_{600}$ of 1.0 mL subcultures in cuvettes. To calculate the colony forming unit / mL (CFU / mL), a cell viability assay was performed with serial dilutions in 1.5mL sterile conical tubes (Eppendorf; Hamburg, Germany). Each dilution ([Table 1]) factor ([Fig 2]) for the respective time points were plated in triplicate on nutrient agar plates by adding 100 μL to the center of the plate and aseptically dispersed using a sterile spreader. Inoculated plates were incubated, inverted, at 37˚C overnight. Colonies were counted using a Sphere Flash® Automated Colony Counter (Neutec Group Inc, NY). A calibration curve was plotted as subculture incubation time to $OD_{600}$, subculture incubation time to CFU/mL, and $OD_{600}$ to CFU/mL. The $R^2$ value ensured the linear fit of $OD_{600}$ to CFU/mL.

## DNA extraction

Using 100 mL of sterile NB, overnight culture was inoculated with a sterile inoculating loop. Fifty milliliters of sterile NB subculture were inoculated with 100 μL of the overnight culture and incubated in a shaking incubator at 37˚C, 150 rpm. The incubation was monitored until a $OD_{600}$ reading of ~ 0.866 nm (~ $4.53 \times 10^8$ CFU / mL). Aliquots of subcultures (1.0ml) were inoculated into six, 1.5 mL sterile cryogenic tubes and centrifuged at 2000 rpm for 10 minutes. The supernatant was decanted, minimizing disturbance to the pelleted cells. DNA was extracted from the pelleted cells using the QIAgen DNeasy Blood and Tissue Kit protocol and QIAgen Qiacube (QIAGEN, Germany). The DNA concentration of extracts were measured, spectrophotometrically, using a NanoDrop™ One/One$^C$ Microvolume UV-Vis Spectrophotometer (Thermo-Fisher Scientific, MA).

## Inoculation and harvesting of pipette tips

Genetic material was inoculated into all pipette tips by aspirating and dispensing 8.0 μL of the medium with extracted DNA. The inoculated tips were re-racked into the pipette box and incubated at room temperature under a BSL-2 biosafety cabinet for 20 minutes. The inoculated

**Table 1. Collection of plated dilution factors for time points.**

| Time (hr) | 0 | 1 | 3 | 5 | 6 | 7 | 8 | 9 |
|---|---|---|---|---|---|---|---|---|
| Dilution Factor ($X^{-1}$) | 3 | 1, 2, 3 | 3, 4, 5 | 5, 6, 7, 8, 9 | 5, 6, 7, 8, 9 | 5, 6, 7, 8, 9 | 5, 6, 7, 8, 9 | 5, 6, 7, 8, 9 |

Table 1 showed the calibration curve dilution factor plating scheme with 32 dilution factors plated in total.

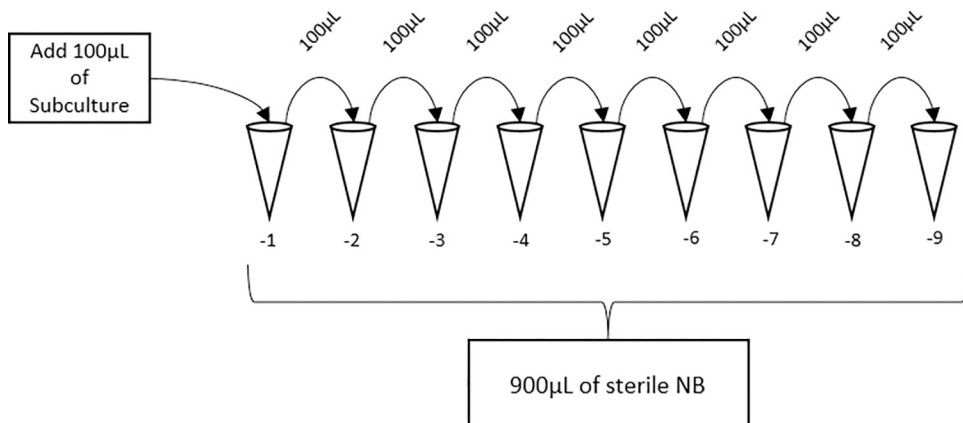

**Fig 2. Serial dilution.** Fig 2 showed a schematic of a serial dilution methods for *A. hydrophila*, in which original 100 μL of subculture was diluted into 900 μL of sterile NB. This process was repeated to the desired dilution factor.

pipette tips were subjected to streamlined decontamination. Residual genetic material was harvested from exposed pipette tips.

A power test was performed to determine a sample size (N = 63) that would indicate significant difference between control and experimental groups. Assuming binary outcomes, $p_0$ of the control group had a "True" outcome, $p_1$ of the exposed group had a "True" outcome, whether $p_0 = p_1$ was tested.

$p_0$ = number of "True outcome" in Control group / total number of samples tested in Control group

$p_1$ = number of "True outcome" in Experimental group / total number of samples tested in Experimental group

The two-sample test were performed (Fig 3), as the power test method for statistical significance. Two separate power tests were conducted for—(1.) positive control vs exposed to DNA

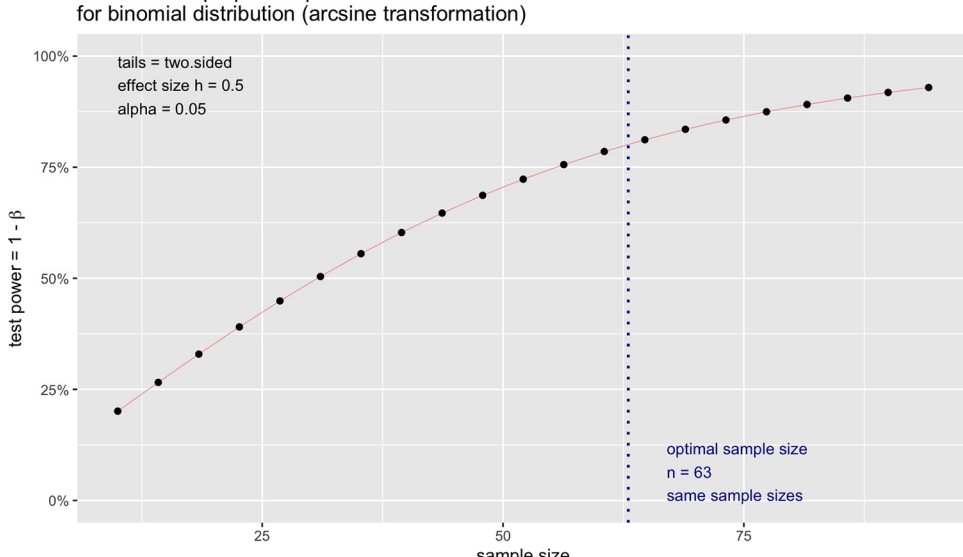

**Fig 3. Power test.** Fig 3 showed the power test for statistical validation, in which sample number on X axis and power for detectable difference on Y axis. Hence, the optimal sample number was chosen as 63.

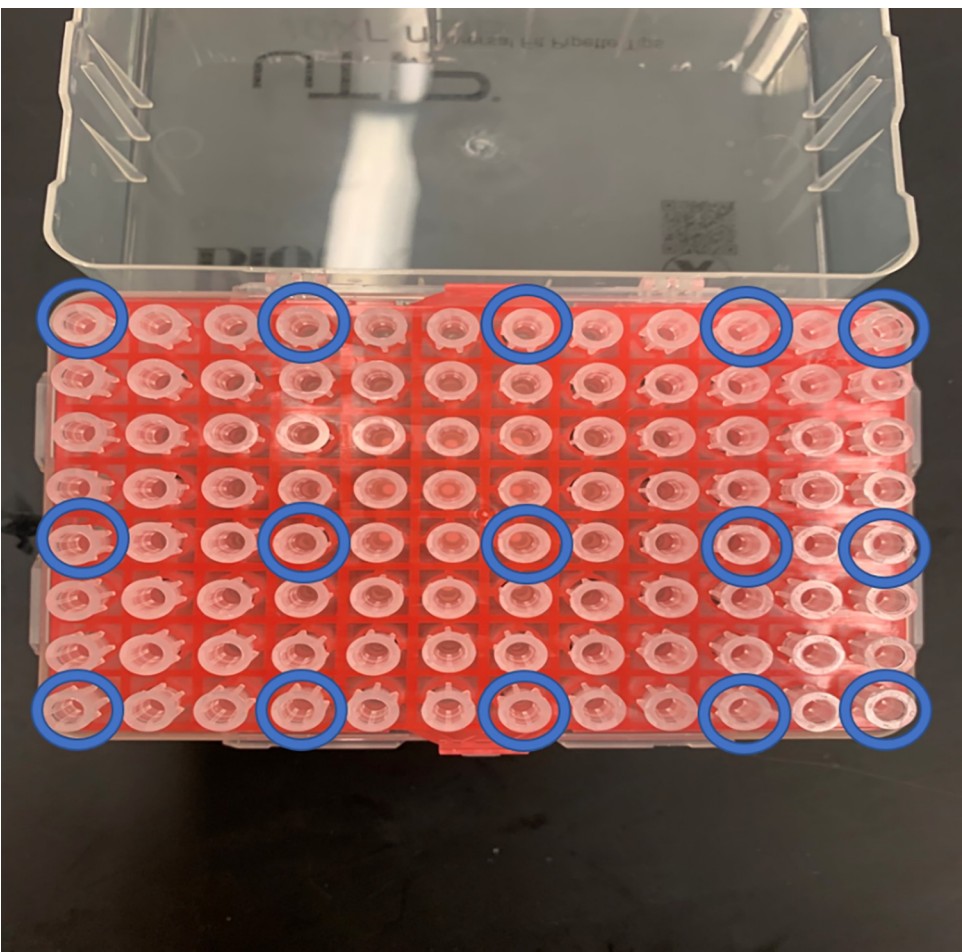

**Fig 4. Tip selection.** Fig 4 showed the diagram of pipette tip selection, in which five samples were chosen from the first, fifth, and last row. The five samples were at the first, fourth, seventh, tenth, and last column. The experimental samples were circled in blue.

and (2.) negative control vs exposed to DNA. The parameters for this power test included significance levels (p values) of 0.05, 0.10, 0.15, power of 0.8, effect sizes (conventional effect size for the two-sample test for proportions) of medium effect (h = 0.5) and was a two-sided test. The resulting sample size was n = 63 (individual pipette tips tested). From five pipette tip boxes, at least 13 pipette tips from each box (Fig 4) were chosen as control or experimental groups. To demonstrate the effect of decontamination, tips were sampled evenly from the conditioned box.

Eight microliters of DNase/RNase free water were added into each well of a sterile 96 well plate. The conditioned tips (tips with DNA harvested from *A. hydrophila* were used to respectively aspirate and dispense 8 μL of DNase/RNase free water in each well thrice. Sterile pipette tips (negative controls) were used to aspirate 3 μL from each well.

### Laboratory detergent preparation

The detergent / bleach wash solution was prepared (Fig 5) in two 6 L Erlenmeyer flasks filled with 2L of deionized water (DIH$_2$O), each. A magnetic stir rod was added to one of the two flasks along with 100g of Alconox® powder (Alconox, NY). This flask was placed on a Nuova

**Alconox/Bleach Wash Solution**

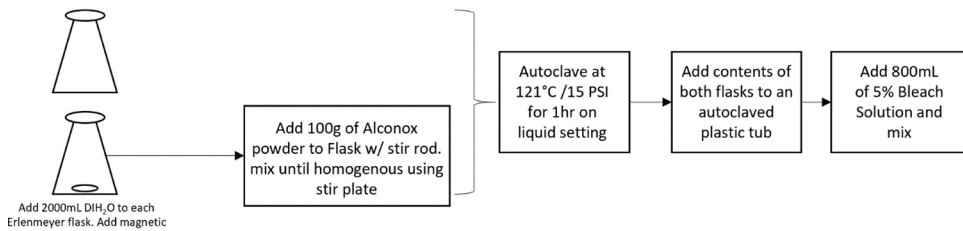

**Fig 5. Wash solution.** Fig 5 showed preparation methods of detergent bath with Alconox/ Bleach wash solution. Two liters of deionized water are added into two Erlenmeyer flasks with a magnetic stir rod each. Then, 100g of Alconox powder was mixed on a stir plate. Both flasks were placed in an autoclave safe bin with one inch of water and autoclaved at 121˚C, 15 psi for one hour. The contents of both flasks were well mixed with 800 mL of 5% Bleach solution in a sterile autoclave safe bin.

Stir Plate (Thermo Fisher Scientific, MA) and mixed until fully dissolved. Both flasks were covered with aluminum foil, placed in an autoclave safe bin with one inch of water and autoclaved at 121˚C, 15 psi for one hour (Modular Component Systems LLC, MD). Following this step, autoclavable tubs was cleaned using Alconox® and $DIH_2O$ and autoclaved on a gravity setting at 121˚C, 15 psi for 45 minutes. The contents of both flasks were added to sterilized/autoclaved tubs and mixed gently. Lastly, 800 mL of 5% bleach solution was added to the solution. The final concentration of the wash solution was 2.5% Alconox® and 1% bleach.

## Wash protocol

The workflow using the detergent /bleach wash solution (Fig 6) is indicated. Five boxes of sterile tips were allocated for each respective testing group. Groups A and B were exposed to the DNA extracts, while Group C remained sterile/unexposed to genetic material and was placed into the corresponding bath. Packaged sterile pipette tip boxes were cleaned with 70% ethanol

**Alconox/Bleach Wash Protocol**

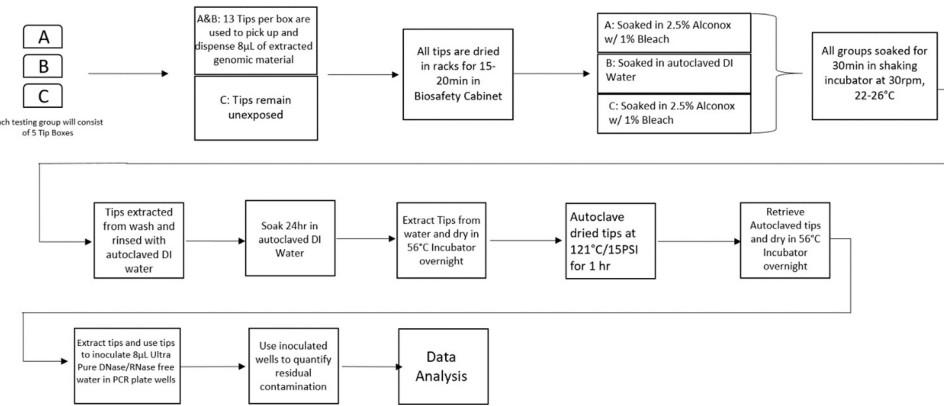

**Fig 6. Wash protocol.** Fig 6 showed the workflow of the tip wash protocol. Among 3 test groups of A, B, and C with 5 sterile tips boxes each, C remained unexposed while A and B were used to draw and dispense 8 μL of extracted genomic material. After 15–20 minutes incubation in a Biosafety cabinet, A and C were soaked in 2.5% Alconox/ Bleach solution, while B was exposed to deionized water only. All groups were soaked for 30 minutes in shaking incubator at 30 RPM and 22–26˚C. All groups were washed with autoclaved deionized water for 24 hours and dried in 56˚C incubator overnight. Dried tips were autoclaved at 121˚C and 15 PSI for 1 hour and dried in 56˚C incubator overnight. Dried tips were used to uptake and dispense 8 μL of DNase/RNase free water for absolute quantification via qRT–PCR.

and placed in a BLS-2 biosafety cabinet. Post-inoculation, tips, tip racks, and their boxes were separately placed into their corresponding wash solutions (A/B/C). The racks floated on top of the liquid with the exposed portion of the tips fully submerged. The wash solutions containing the tip racks were placed into a shaking incubator set at room temperature (25°C) at 30 rpm and soaked for 30 minutes. The tip racks and boxes were removed from wash solution and rinsed with autoclaved DI $H_2O$. The pipette tip boxes were filled with 250 mL of autoclaved $DIH_2O$, and the tip racks were placed back in the box such that the tip-ends were fully submerged in the water for 24 hours. After the incubation, the boxes were dehydrated, and tip racks were re-racked in their respective boxes. These boxes were placed into a 56°C incubator for 24 hours with the lid slightly ajar for ventilation. The pipette tip boxes were autoclaved for 1 hour at 121°C, 15 PSI (5 minutes of dry time) on a gravity setting. Retrieved autoclaved samples were placed back in the 56°C incubator for 24 hours to ensure dehydration. Finally, the harvested medium from tips were evaluated with qRT—PCR.

## Ozone vapor

For ozone decontamination conditions, at least 250 PPM (parts per million) * minute was set for low exposure and at least 14400 PPM * minute, for high [14] exposure. An ozone decontamination complex (ODC) was streamlined (Fig 7) with an Airthereal B50 Mini ozone generator (Sain Store, NV), a portable small cool mist humidifier (Geniani), and a pump (Mambate USA, NY) with connecting tubing packed with Carulite 200 (Oxygen Technologies, Canada) in an enclosed, plastic chamber. The ODC was kept under a fume hood to mitigate any potential leakage. Respective pipettes were inoculated, cleaned, and processed accordingly; a

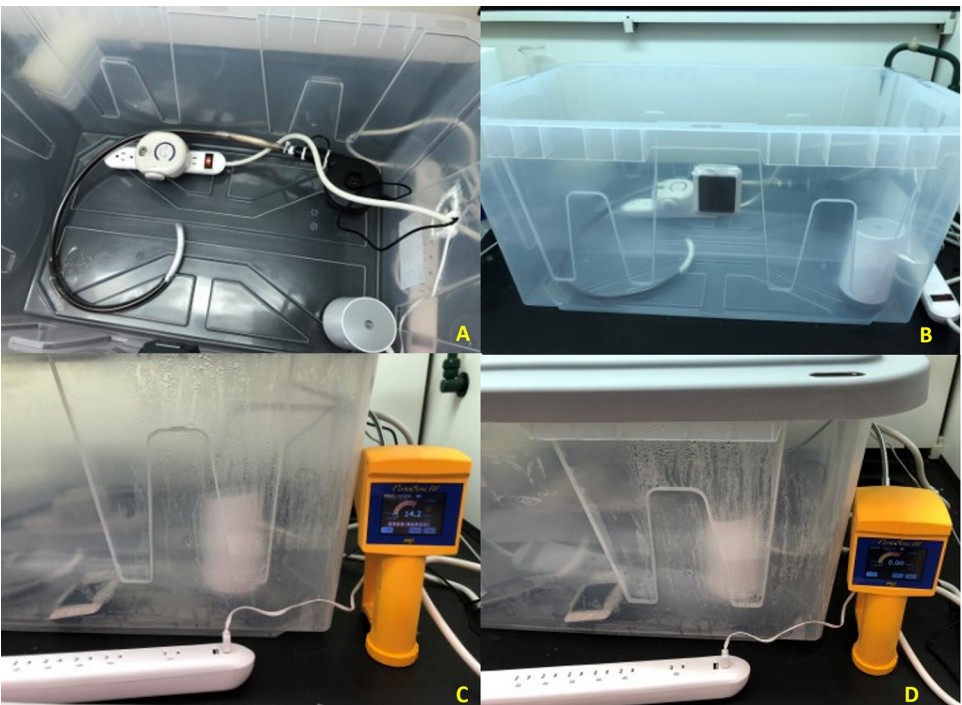

**Fig 7. Ozone decontamination chamber.** Fig 7 showed pictures of A) Top view of the ODC from left are Carulite 200 packed tubing using glass wool, ozone generator, pump with tubing, and humidifier, (B) Side view of the ODC, (C) Side view of the ODC during exposure, ozone sensor: 14.2 PPM (parts per million) and (D) Side view of the ODC post neutralization, ozone sensor: 0.0 PPM.

humidifier maintained 80% humidity. Samples were placed in the ODC chamber and sealed. Exposure treatments were 30 minutes or 24 hours 10 PPM of ozone vapor. After exposure, the ozone generator was turned off and the pump along with tubing filled with the neutralizing catalyst were turned on. When the ozone level reached 10 PPM for 5 minutes, the experimental samples were extracted from the chamber and incubated at 56°C overnight to ensure dehydration. Post-ozone exposure, samples were tested by qPCR for residual genetic material.

## Cold atmospheric plasma exposure

The operating parameters of CAP instrumentation was set as follows: input voltage of 25 V, regulated voltage of 12 V, discharge voltage of 6.5 kV, frequency of 12.5 kHz, and helium gas flowrate range of 5 LPM (liters per minute). Under a stabilized plasma jet stream (Fig 8), the pipette box with conditioned pipette tips were placed 2 to 5 cm under the plasma jet. Parameters tested were 1 minute upright exposure and 1 minute inverted exposure. The treated tips were incubated at 56°C overnight to ensure dehydration. Similar to other decontamination methods described (Fig 9), following CAP treatment samples were harvested and tested by qRT—PCR.

## Primer design and validation

The initial primer design for qRT–PCR were based [18] on the literature. The housekeeping gene, 16S rRNA, was selected for *A. hydrophila*, for primer stability [16] and long half-life. Primers (Table 2) were validated using NCBI's Primer BLAST and the Integrated DNA Technologies (IDT) Oligoanalyzer tools. Criteria for primer selection included the following: specifically to documented target sequences, amplicon size for qPCR analysis, juxtaposing annealing temperature, and acceptable ΔG values.

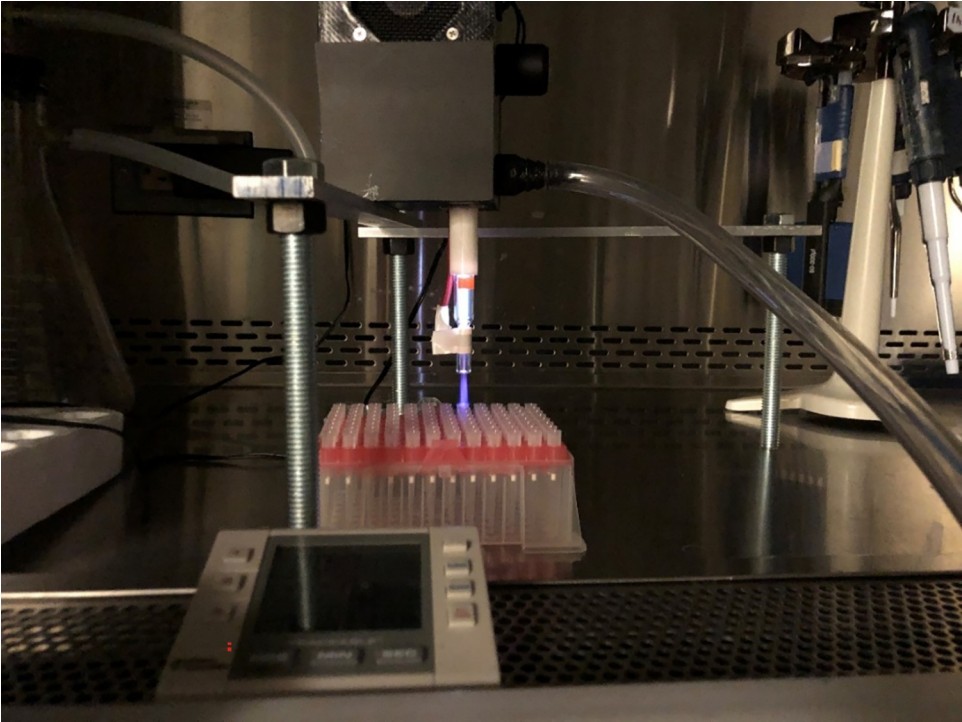

**Fig 8. Cold atmospheric plasma.** Fig 8 showed the layout of pipette tips during CAP exposure. Plasma jet was 2–5 cm from the upright position or tips (as shown above) or inverted (tip side closest to the jet).

# *Ozone / CAP Protocol*

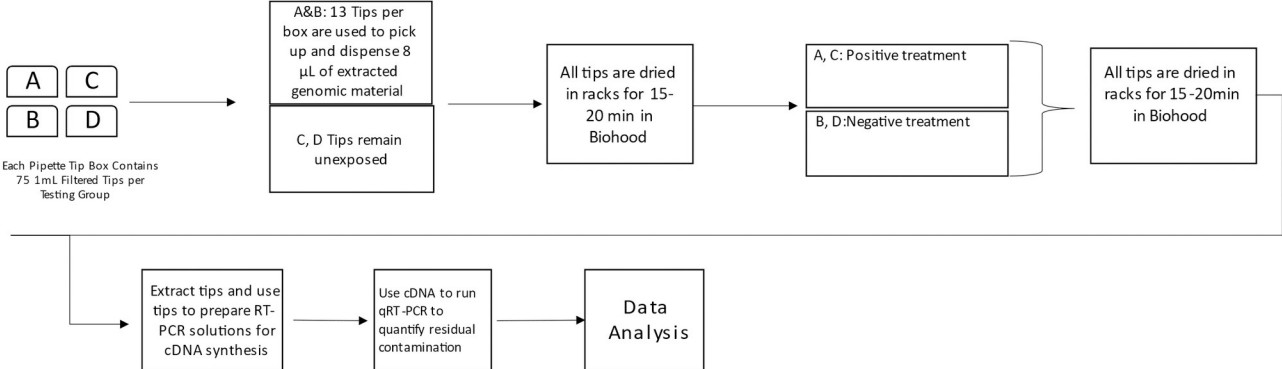

**Fig 9. Ozone and CAP protocol.** Fig 9 showed the workflow of ozone / CAP decontamination protocol. Among 4 test groups of A, B, C, and D with 5 sterile tips boxes each, C and D remained unexposed while A and B were used to draw and dispense 8 μL of extracted genomic material. After 15–20 minutes incubation in a Biosafety cabinet, A and C were exposed to positive treatment of decontamination, while B and D were remained untreated. All groups were dried in 15–20 minutes in BSL—2 biosafety cabinet. Dried tips were used to uptake and dispense 8 μL of DNase/RNase free water for absolute quantification via qRT–PCR.

## qRT—PCR data processing validation

Based on "CDC 2019 Novel Coronavirus (nCoV) Real-Time RT-PCR Diagnostic Panel—Instructions for Use", the limit of detection for samples with 2019-nCoV_N1 initial RNA concentration of $10^{0.5}$ RNA copies / μL (~ 3162 copies / mL) was reported, and a mean threshold cycle value of 32 with 100% positive detection test results (20 / 20; Positive / Total). Samples with 2019-nCoV_N1 initial RNA concentration of $10^0$ RNA copies / μL (~ 1000 copies / mL) reported the mean threshold cycle value of 32.8 with 100% positive detection test [19] results (20 / 20; Positive / Total). Based on these values, the threshold number of DNA copies / mL present in a detectable sample was calculated with the following Eq (1): (assuming 100% efficiency for PCR reactions)

$$DetectCT = I \times 2^{C_t} \tag{1}$$

DetectCT = The threshold number of DNA copies / mL present in a detectable sample
I = Initial amount of DNA copies / mL
$C_t$ = Mean threshold cycle
Therefore, the threshold number of DNA copies / mL present in a detectable sample was 1.358 X $10^{13}$ RNA copies / mL. (~7.478 X $10^{12}$ DNA copies / mL based on the second set of data.)

**Table 2. Primer validation for *A. hydrophila*.**

| Target Gene | FP/RP | Primer Sequence (5'-3') | Primer BLAST | | IDT Oligoanalyzer | | | Source |
|---|---|---|---|---|---|---|---|---|
| | | | Tm (°C) | GC% | Hairpin ΔG (kCal/mol) | Self-Dimer ΔG (kCal/mol) | Hetero-Dimer ΔG (kCal/mol) | |
| 16s rRNA | FP | GCGGCGGACGGGTGAGTA | 64.44 | 72.22 | -1.58 | -3.61~-0.96 | -11.09~-1.6 | [18] |
| | RP | CCCACTGCTGCCTCCCGT | 64.41 | 72.22 | | | | |

Table 2 showed the primer validation: this table was organized left to right with the following categories: Target Gene, Forward Primer (FP)/Reverse Primer (RP), Primer Sequence (from 5' to 3'), Melting Temperature (Tm), G/C content %, Amplicon Size, Hairpin Value, Self-Dimer Value, Hetero-Dimer Value, and Literature Review Citation.

The expected mean threshold cycle for a positive sample with known initial amount of RNA is calculated with the following Eq (2):

$$C_t = \left( \frac{DetectCT}{I} \right) \tag{2}$$

$C_t$ = Expected mean threshold cycle

Detect CT = The threshold number of DNA copies / mL present in a detectable sample

I = Initial amount of DNA copies / mL

Exposed and sterilized pipette tips that show threshold cycle value smaller than the expected mean threshold cycle will be considered positive test results.

Example Calculation 1

Initial number of copies: $3.544 \times 10^{11}$ DNA copies / mL

Threshold number of DNA copies / mL present in a detectable sample: $1.358 \times 10^{13}$ RNA copies / mL

Expected mean threshold cycle of a detectable sample: 5.26 cycles

These experimental conditions resulted in threshold cycle that were too low.

Example Calculation 2 (1:1000000 dilution of DNA sample)

Initial number of copies: $3.544 \times 10^{5}$ DNA copies / mL

Threshold number of DNA copies / mL present in a detectable sample: $1.358 \times 10^{13}$ RNA copies / mL

Expected mean threshold cycle of a detectable sample: 25.19 cycles

The expected threshold cycle was between 25 to 35 and therefore representative of optimal assays conditions.

The limit of detection was updated to the experimental mean threshold cycle value of negative controls (uninoculated and untreated).

## qRT—PCR procedures

qRT–PCR was used to quantify genetic material. Methods for qRT-PCR were according to manufacturer instructions: 5.0 µL of SYBR green master mix and 1.0 µL of primer-pairs specific to housekeeping genes, which is normalized to ~500 nM for both forward and reverse sets. Assay conditions on a BioRad CFX 96 Real-Time Thermocycler (Biorad; CA) were: heat-activation 50˚C for 10 minutes, 5 minutes at 95˚C, 50 cycles of denaturation at 95˚C for 5 seconds and annealing/extension at 53˚C for 30 seconds. This protocol was repeated at three different annealing temperatures to identify optimal conditions, using BioRad CFX96. Reaction efficiencies were determined by comparing the slope of the calibration curve. The non-template control (NTC) threshold cycles for both reverse and forward primers were compared to determine the annealing temperature and primer pairs that optimize reaction efficiency as well as the quantification limit, or lower limit of quantification (LLOQ) for the assay. A $C_t$ value that was above the NTC $C_t$ was considered indeterminate.

## Turnover ratio

To compare different decontamination effects, turnover ratio was defined and compared as follows (3):

$$TO = \frac{N_{Rep}}{N_{Tot}} \tag{3}$$

TO = Turnover ratio

$N_{Rep}$ = Number of samples considered as decontaminated

$N_{Tot}$ = Total sample number

For qRT—PCR results, decontaminated results accounted for number of samples that have mean threshold cycle larger than the limit of detection.

## Log reduction

To compare different decontamination effects, log reduction was defined as the following **(4)**:

$$LR = (C_{Ino}) - (C_{Har}) \tag{4}$$

LR = Log reduction

$C_{Ino}$ = Concentration of inoculated

$C_{Har}$ = Concentration of harvested

The concentrations of genetic material were calculated from the qPCR mean threshold cycle values and the following Eq (5):

$$C_{DNA} = \frac{DetectCT}{2^{C_t}} \tag{5}$$

$C_{DNA}$ = DNA concentration

DetectCT = Threshold number of DNA copies / mL in a detectable sample

$C_t$ = Mean threshold cycles

## Statistical analysis

The log reduction for each experimental condition were analyzed using one-way ANOVA at a p-value of 0.05, followed by a Tukey's post hoc test to compare the means of all experimental conditions to the mean of each experimental condition. GraphPad Prism (version 9.0.0) and R (version 4.2.1) were used to generate graphs and perform statistical analyses.

## Results

### Bacterial growth calibration curve

The calibration curves (Fig 10–12) displayed normal bacterial growth patterns: lag phase proceeding inoculation (T0-T3), followed by a log phase of growth (T4-8), and finally a plateau (T9). The CFU/mL /OD$_{600}$ displayed a linear growth progression ($R^2$ = 0.923) indicative of standard bacterial growth.

### DNA extraction results

Seven aliquots of subculture were extracted at OD$_{600}$ of 0.823. These aliquots (Table 3) were processed as discussed previously and the concentration was determined spectrophotometrically, by Nanodrop (Thermo-Fisher Scientific, MA).

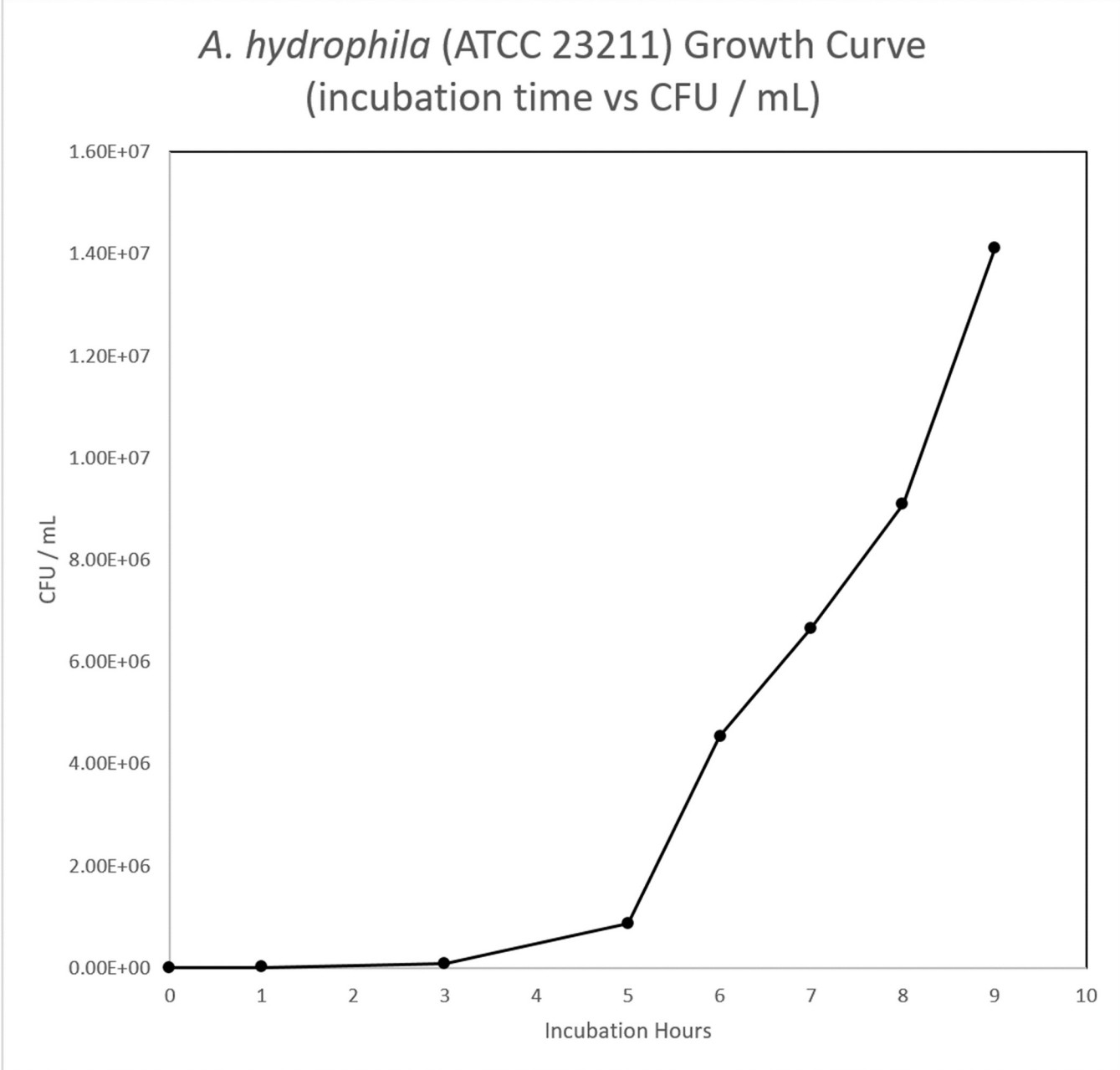

**Fig 10. *A. hydrophila* growth curve time vs CFU/mL.** Fig 10 showed Time vs CFU/ mL readings to measure *A. hydrophila* (ATCC 23211) demonstrated bacterial growth at lag and log phases.

### Processed qRT—PCR data

Turnover ratio and log reduction (Table 4) were used to compare the efficacy of laboratory detergent, ozone vapor, and CAP. Detergent showed the highest turnover ratio and log reduction, followed by ozone vapor and CAP.

### Log reduction graph

This log reduction graph (Fig 13) provided a visual representation of the log reduction values extrapolated from the qRT–PCR threshold cycle results. Negative log reduction on positive

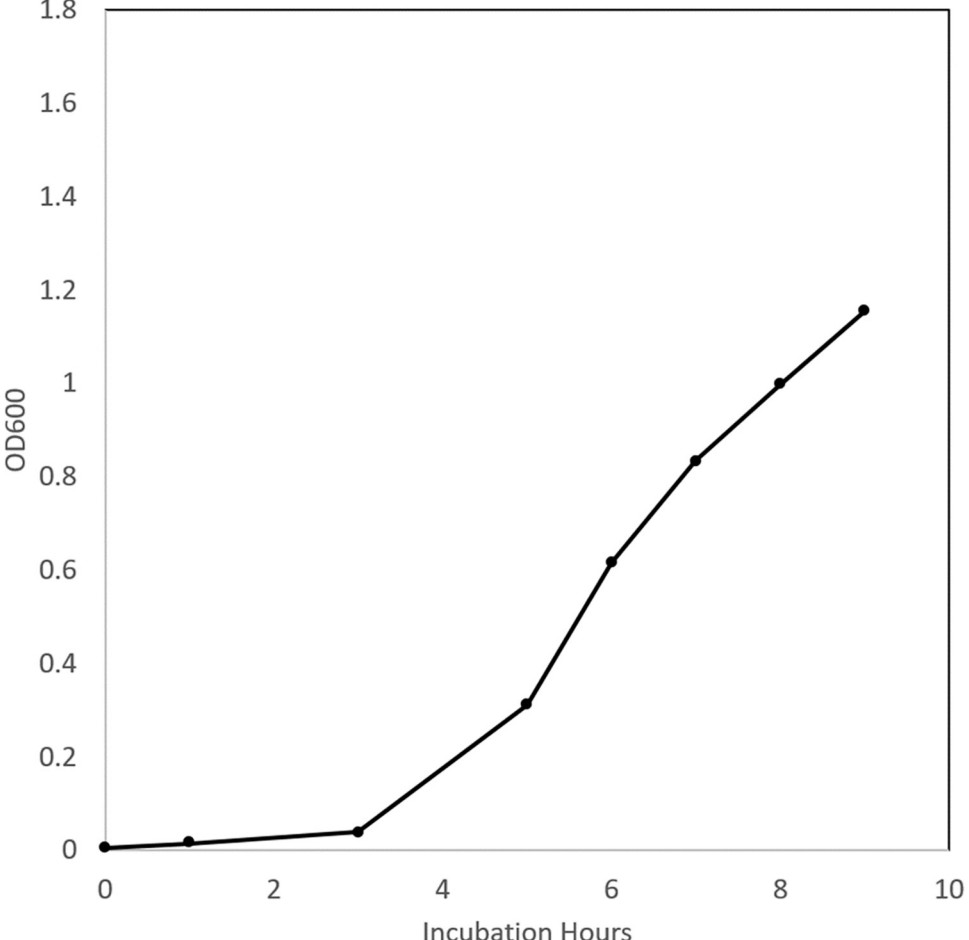

**Fig 11. *A. hydrophila* growth curve time vs OD$_{600}$.** Fig 11 showed Time vs OD$_{600}$ readings to measure *A. hydrophila* (ATCC 23211) demonstrated bacterial growth at lag and log phases.

control signified presence of replicable genetic material when compared to negative control without any genetic material. For Detergent Control, Ozone Control, and CAP Control, the average concentration of negative control was the inoculated concentration. For experimental groups other than Detergent Control, Ozone Control, and CAP Control, the average concentration of positive control was the inoculated DNA concentration.

## Selected statistical results

One-way ANOVA test indicated there was a statistically significant difference (p-value < 0.001) among log reduction in Positive Control, Negative Control, Detergent 2.5%, Detergent Control, Ozone 250 PPM*min, Ozone 14400 PPM*min, Ozone Control, CAP Upright, CAP Inverted, and CAP Control. Tukey's multiple comparisons of the log reduction difference between all experimental conditions to every other experimental condition were calculated (Table 5). Differences were considered statistically significant when p-value is <0.05 and the

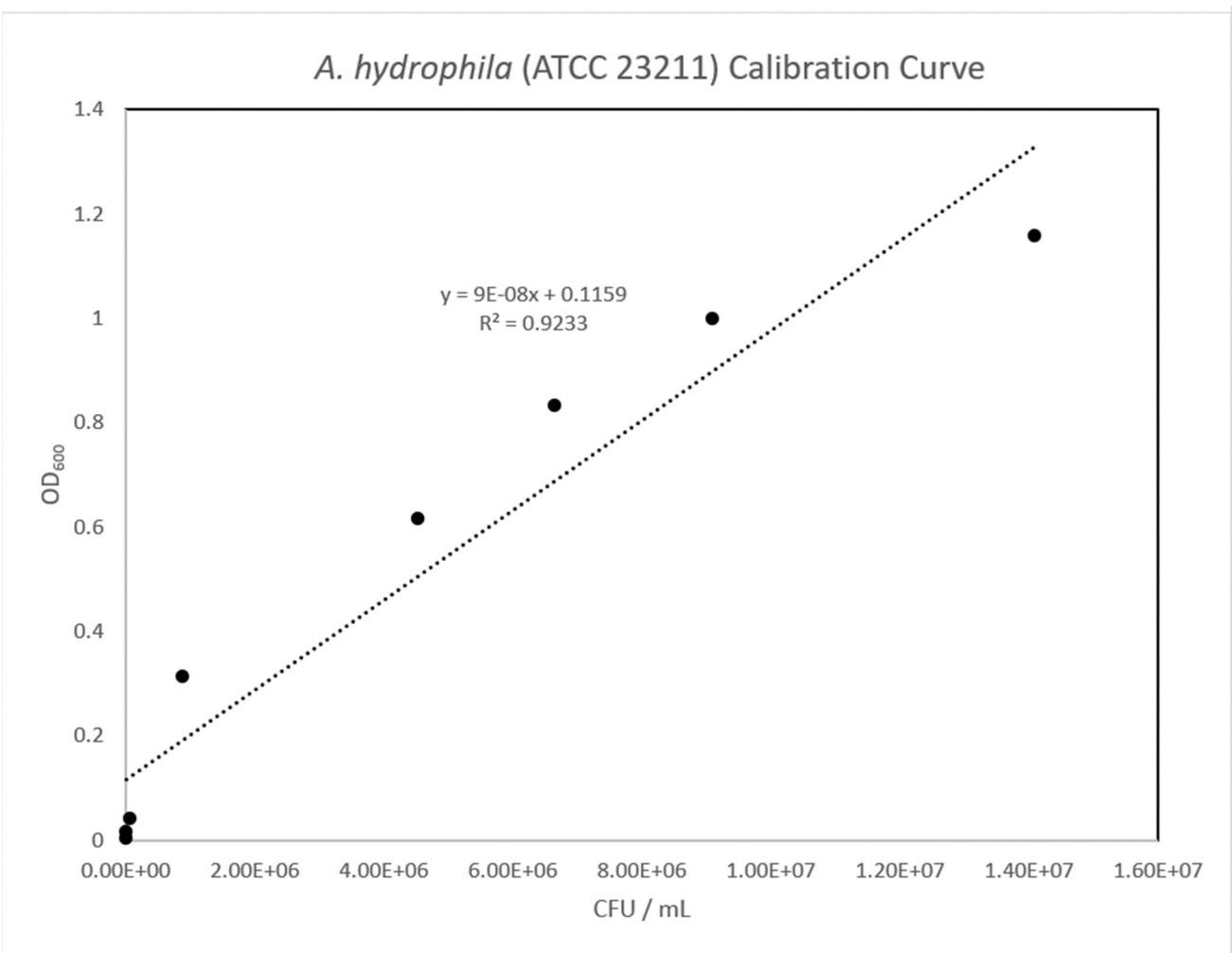

**Fig 12. *A. hydrophila* calibration curve.** Fig 12 showed a calibration curve with CFU/ mL on X axis and $OD_{600}$ on Y axis. The highly linear correlation was supported by $R^2$ value of 0.92.

p-value was categorized as ** = $p < 0.01$, *** = $p < 0.001$, **** = $p < 0.0001$, and ns = nonsignificant based on level of significance. Non-select statistical results can be found in supplemental materials (Table 6).

**Table 3. *A. hydrophila* DNA extraction results.**

| Sample Name | Concentration (ng/μL) | Absorbance ratio 260/280 | Absorbance ratio 260/230 | Absorbance 260 (nm) | Absorbance 280 (nm) |
|---|---|---|---|---|---|
| Sample 1 | 109.5 | 2.12 | 1.80 | 2.19 | 1.03 |
| Sample 2 | 110.3 | 2.14 | 1.68 | 2.21 | 1.03 |
| Sample 3 | 121.1 | 2.14 | 1.75 | 2.42 | 1.13 |
| Sample 4 | 125.9 | 2.14 | 1.86 | 2.52 | 1.18 |
| Sample 5 | 145.9 | 2.13 | 2.04 | 2.92 | 1.37 |
| Sample 6 | 136.6 | 2.13 | 1.97 | 2.73 | 1.28 |

Table 3 showed DNA concentrations (ng/μL) and absorbance values (nm) for extracted *A hydrophila* aliquots.

**Table 4. Results of qRT-PCR analysis of decontaminated pipette tips.**

| | Conditions | | | | | | | | | |
|---|---|---|---|---|---|---|---|---|---|---|
| | Positive Control | Negative Control | Detergent Control | Detergent 2.5% | Ozone Control | Ozone 250 PPM*min | Ozone 14400 PPM*min | CAP Control | CAP Upright | CAP Inverted |
| Average Threshold Cycle | 16.316 | 30.213 | 37.130 | 36.058 | 31.039 | 27.440 | 31.301 | 31.439 | 25.047 | 29.611 |
| Cleaned Sample | 0 | 36 | 71 | 70 | 58 | 13 | 62 | 59 | 9 | 43 |
| Total Sample Number | 63 | 63 | 73 | 73 | 63 | 63 | 63 | 63 | 63 | 63 |
| Turnover Ratio | 0.000 | 0.571 | 0.973 | 0.959 | 0.921 | 0.206 | 0.984 | 0.937 | 0.143 | 0.683 |
| Log Reduction | -4.183 | 0.000 | 2.082 | 5.943 | 0.248 | 3.349 | 4.511 | 0.369 | 2.628 | 4.002 |
| Threshold Cycle Standard Deviation | 2.093 | 1.080 | 5.659 | 4.679 | 1.198 | 4.145 | 0.660 | 1.302 | 4.236 | 2.467 |
| Percent Error | 0.128 | 0.036 | 0.152 | 0.130 | 0.039 | 0.151 | 0.021 | 0.041 | 0.169 | 0.083 |
| Error Bar (Log) | 0.537 | 0.000 | 0.317 | 0.771 | 0.010 | 0.506 | 0.095 | 0.015 | 0.445 | 0.333 |
| Processed Error Bar (Log) | 0.537 | 0.009 | 0.317 | 1.308 | 0.010 | 1.042 | 0.632 | 0.015 | 0.981 | 0.870 |

Table 4 showed the results of qRT-PCR Analysis Following Decontamination of Pipette Tips: this table was organized left to right with the following experimental conditions: Positive Control (DNA exposure only), Negative Control (clean tip), Detergent Control (clean tip exposed to Detergent only), Detergent 2.5% (DNA exposed then Detergent treated), Ozone Control (clean tip exposed to Ozone only), Ozone 250 PPM*min and Ozone 14400 PPM*min (DNA exposed then Ozone treated accordingly), CAP Control (clean tip exposed to CAP only), CAP Upright and Inverted (DNA exposed then CAP treated accordingly). The rows referred to categories: average threshold cycle, cleaned sample number, total sample number, turnover ratio, log reduction, threshold cycle standard deviation, percent error, processed error bar (log reduction graph).

* Additional samples were tested due to reaction condition variability.

**Table 5. Selected statistical results.**

| Tukey's multiple comparisons test | Mean Diff. | 95.00% CI of diff. | Summary | Adjusted P Value |
|---|---|---|---|---|
| Negative Control vs. Detergent Control | -2.082 | -2.484 to -1.680 | **** | <0.0001 |
| Negative Control vs. Ozone Control | -0.248 | -0.6645 to 0.1685 | ns | 0.6757 |
| Negative Control vs. CAP Control | -0.369 | -0.7855 to 0.04754 | ns | 0.1343 |
| Detergent 2.5% vs. Ozone 250 PPM*min | 2.594 | 2.192 to 2.996 | **** | <0.0001 |
| Detergent 2.5% vs. Ozone 14400 PPM*min | 1.432 | 1.030 to 1.834 | **** | <0.0001 |
| Detergent 2.5% vs. CAP Upright | 3.315 | 2.913 to 3.717 | **** | <0.0001 |
| Detergent 2.5% vs. CAP Inverted | 1.941 | 1.539 to 2.343 | **** | <0.0001 |
| Detergent Control vs. Ozone Control | 1.834 | 1.432 to 2.236 | **** | <0.0001 |
| Detergent Control vs. CAP Control | 1.713 | 1.311 to 2.115 | **** | <0.0001 |
| Ozone 250 PPM*min vs. Ozone 14400 PPM*min | -1.162 | -1.579 to -0.7455 | **** | <0.0001 |
| Ozone 250 PPM*min vs. CAP Upright | 0.721 | 0.3045 to 1.138 | **** | <0.0001 |
| Ozone 250 PPM*min vs. CAP Inverted | -0.653 | -1.070 to -0.2365 | **** | <0.0001 |
| Ozone 14400 PPM*min vs. CAP Upright | 1.883 | 1.466 to 2.300 | **** | <0.0001 |
| Ozone 14400 PPM*min vs. CAP Inverted | 0.509 | 0.09246 to 0.9255 | ** | 0.0045 |
| Ozone Control vs. CAP Control | -0.121 | -0.5375 to 0.2955 | ns | 0.9957 |
| CAP Upright vs. CAP Inverted | -1.374 | -1.791 to -0.9575 | **** | <0.0001 |

Table 5 shows Tukey's multiple comparisons of the Log Reduction difference between all experimental and control conditions.

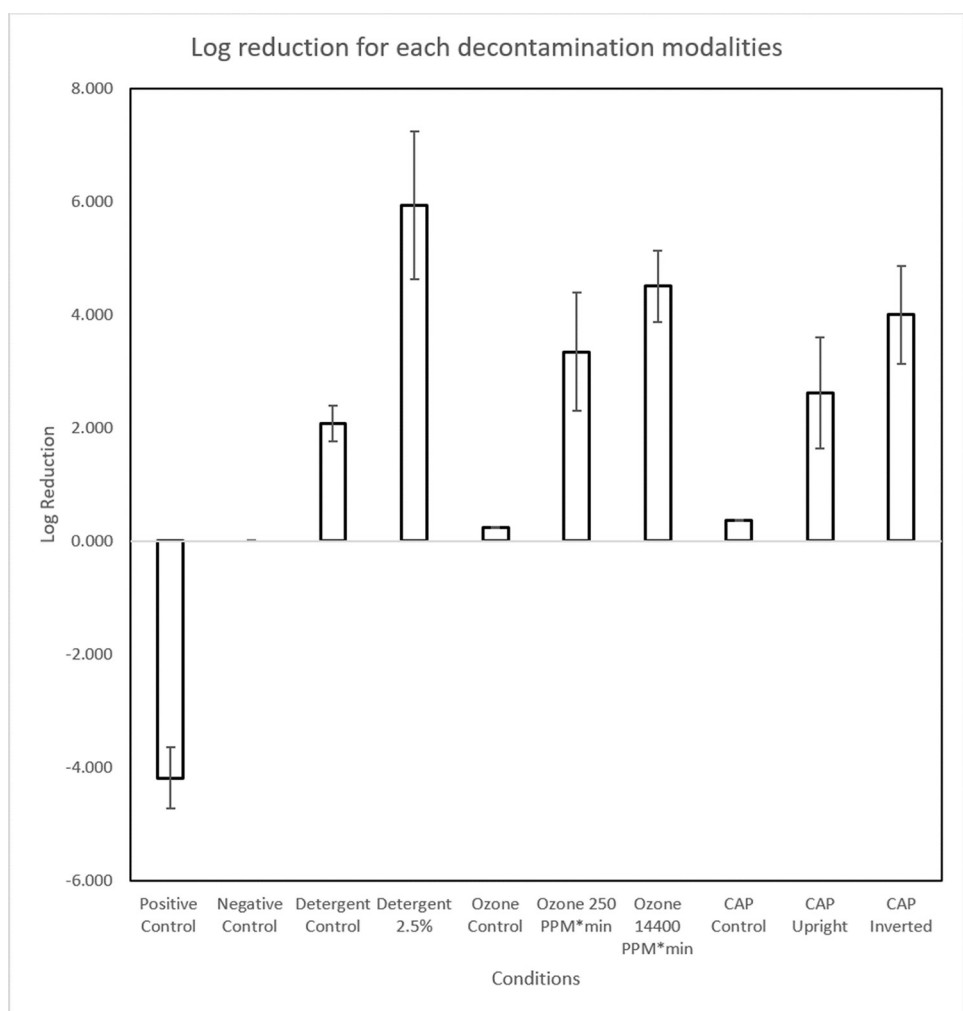

**Fig 13. Log reduction.** Fig 13 showed log reduction of genomic material post decontamination compared with controls on Y axis for different test conditions on X axis. The error bars calculated based on the standard deviation of experimental groups. The log reduction of positive and negative control was -4.183 ± 0.537 and 0 ± 0.009, respectively. Those of detergent control and 2.5% were 2.082 ± 0.317 and 5.943 ± 1.308, respectively. Those of ozone control, 250 PPM*min, and 14400 PPM*min were 0.249 ± 0.010, 3.349 ± 1.043, and 4.511 ± 0.632, respectively. Those of CAP control, upright, and inverted were 0.369 ± 0.015, 2.628 ± 0.981, and 4.002 ± 0.870, respectively.

## Discussion

The objective of this study was to identify decontamination methods for plasticware (pipette tips) that would permit re-use, and specifically testing that may require handling genetic materials. The goal was to measure residual genetic materials following decontamination and provide insight into methods that may prove resourceful during emergencies (i.e., pandemic/endemic scenarios with scarcity of supplies). Compared with positive and negative controls, the results demonstrated that DNA residue, following decontamination, can interfere with qRT—PCR results.

All methods were shown to have decontamination efficacies. Lab detergent produced the highest log reduction (5.943) in residual genetic material, however the excessive residue from the detergent in pipette tips and inner filters could affect downstream assays. The statistically significant difference in detergent controls to negative controls bolstered excessive residue on

**Table 6. Non-select statistical results.**

| Tukey's multiple comparisons test | Mean Diff. | 95.00% CI of diff. | Summary | Adjusted P Value |
|---|---|---|---|---|
| Positive Control vs. Negative Control | -4.183 | -4.600 to -3.766 | **** | <0.0001 |
| Positive Control vs. Detergent 2.5% | -10.13 | -10.53 to -9.724 | **** | <0.0001 |
| Positive Control vs. Detergent Control | -6.265 | -6.667 to -5.863 | **** | <0.0001 |
| Positive Control vs. Ozone 250 PPM*min | -7.532 | -7.949 to -7.115 | **** | <0.0001 |
| Positive Control vs. Ozone 14400 PPM*min | -8.694 | -9.111 to -8.277 | **** | <0.0001 |
| Positive Control vs. Ozone Control | -4.431 | -4.848 to -4.014 | **** | <0.0001 |
| Positive Control vs. CAP Upright | -6.811 | -7.228 to -6.394 | **** | <0.0001 |
| Positive Control vs. CAP Inverted | -8.185 | -8.602 to -7.768 | **** | <0.0001 |
| Positive Control vs. CAP Control | -4.552 | -4.969 to -4.135 | **** | <0.0001 |
| Negative Control vs. Detergent 2.5% | -5.943 | -6.345 to -5.541 | **** | <0.0001 |
| Negative Control vs. Detergent Control | -2.082 | -2.484 to -1.680 | **** | <0.0001 |
| Negative Control vs. Ozone 250 PPM*min | -3.349 | -3.766 to -2.932 | **** | <0.0001 |
| Negative Control vs. Ozone 14400 PPM*min | -4.511 | -4.928 to -4.094 | **** | <0.0001 |
| Negative Control vs. Ozone Control | -0.248 | -0.6645 to 0.1685 | ns | 0.6757 |
| Negative Control vs. CAP Upright | -2.628 | -3.045 to -2.211 | **** | <0.0001 |
| Negative Control vs. CAP Inverted | -4.002 | -4.419 to -3.585 | **** | <0.0001 |
| Negative Control vs. CAP Control | -0.369 | -0.7855 to 0.04754 | ns | 0.1343 |
| Detergent 2.5% vs. Detergent Control | 3.861 | 3.474 to 4.248 | **** | <0.0001 |
| Detergent 2.5% vs. Ozone 250 PPM*min | 2.594 | 2.192 to 2.996 | **** | <0.0001 |
| Detergent 2.5% vs. Ozone 14400 PPM*min | 1.432 | 1.030 to 1.834 | **** | <0.0001 |
| Detergent 2.5% vs. Ozone Control | 5.695 | 5.293 to 6.097 | **** | <0.0001 |
| Detergent 2.5% vs. CAP Upright | 3.315 | 2.913 to 3.717 | **** | <0.0001 |
| Detergent 2.5% vs. CAP Inverted | 1.941 | 1.539 to 2.343 | **** | <0.0001 |
| Detergent 2.5% vs. CAP Control | 5.574 | 5.172 to 5.976 | **** | <0.0001 |
| Detergent Control vs. Ozone 250 PPM*min | -1.267 | -1.669 to -0.8650 | **** | <0.0001 |
| Detergent Control vs. Ozone 14400 PPM*min | -2.429 | -2.831 to -2.027 | **** | <0.0001 |
| Detergent Control vs. Ozone Control | 1.834 | 1.432 to 2.236 | **** | <0.0001 |
| Detergent Control vs. CAP Upright | -0.546 | -0.9480 to -0.1440 | *** | 0.0008 |
| Detergent Control vs. CAP Inverted | -1.92 | -2.322 to -1.518 | **** | <0.0001 |
| Detergent Control vs. CAP Control | 1.713 | 1.311 to 2.115 | **** | <0.0001 |
| Ozone 250 PPM*min vs. Ozone 14400 PPM*min | -1.162 | -1.579 to -0.7455 | **** | <0.0001 |
| Ozone 250 PPM*min vs. Ozone Control | 3.101 | 2.684 to 3.518 | **** | <0.0001 |
| Ozone 250 PPM*min vs. CAP Upright | 0.721 | 0.3045 to 1.138 | **** | <0.0001 |
| Ozone 250 PPM*min vs. CAP Inverted | -0.653 | -1.070 to -0.2365 | **** | <0.0001 |
| Ozone 250 PPM*min vs. CAP Control | 2.98 | 2.563 to 3.397 | **** | <0.0001 |
| Ozone 14400 PPM*min vs. Ozone Control | 4.263 | 3.846 to 4.680 | **** | <0.0001 |
| Ozone 14400 PPM*min vs. CAP Upright | 1.883 | 1.466 to 2.300 | **** | <0.0001 |
| Ozone 14400 PPM*min vs. CAP Inverted | 0.509 | 0.09246 to 0.9255 | ** | 0.0045 |
| Ozone 14400 PPM*min vs. CAP Control | 4.142 | 3.725 to 4.559 | **** | <0.0001 |
| Ozone Control vs. CAP Upright | -2.38 | -2.797 to -1.963 | **** | <0.0001 |
| Ozone Control vs. CAP Inverted | -3.754 | -4.171 to -3.337 | **** | <0.0001 |
| Ozone Control vs. CAP Control | -0.121 | -0.5375 to 0.2955 | ns | 0.9957 |
| CAP Upright vs. CAP Inverted | -1.374 | -1.791 to -0.9575 | **** | <0.0001 |
| CAP Upright vs. CAP Control | 2.259 | 1.842 to 2.676 | **** | <0.0001 |
| CAP Inverted vs. CAP Control | 3.633 | 3.216 to 4.050 | **** | <0.0001 |

Table 6 shows Tukey's multiple comparisons of the Log Reduction difference between all experimental conditions to every other experimental conditions

qRT–PCR results. Based on these results, treatment with lab detergent was not a favorable decontamination method.

The experimental group exposed to ozone vapor at 14400 PPM $*$ minute showed the second highest turnover ratio (98.4%) and log reduction (4.511). Additionally, tips exposed to CAP inverted 1 minute showed the turnover ratio (68.3%) and log reduction (4.002). Ozone vapor and CAP resulted in nondestructive traits for clean pipette tips. The relatively lower turnover ratio and log reduction of CAP could be attributed to the absence of streamlined exposure— the availability of a gantry or similar system could optimize exposure and decontamination. Therefore, CAP exposure could be optimized to develop an efficient method, as the period of exposure and impact to plasticware appear are minimal compared with other decontaminations methods we tested.

## Conclusion

The study concluded that detergent, ozone vapor, and CAP showed different efficacies for genetic material clearance on used pipette tips. Ozone vapor demonstrated the best clearance of genetic material with minimal changes to tip integrity. The level of ozone exposure and humidity should be adjusted according to its application. Optimization of CAP could provide an efficient alternative given the minimal exposure time and preservation of tip integrity.

## Supporting information

**S1 File. Minimal data set for pipette decontamination.**
(XLSX)

## Acknowledgments

The authors would like to acknowledge DBCMS/OSEL/CDRH management for their contributions and use of equipment/materials towards the completion of this study.

**Disclaimer:** The findings and conclusions in this paper have not been formally disseminated by the Food and Drug Administration and should not be construed to represent any agency determination or policy. The mention of commercial products, their sources, or their use in connection with material reported herein is not to be construed as either an actual or implied endorsement of such products by Department of Health and Human Services.

## Author Contributions

**Conceptualization:** Enusha Karunasena.

**Data curation:** Sang Hyuk Lee, William Kastor, Xiao Fu, Vikas Soni.

**Formal analysis:** Sang Hyuk Lee, William Kastor, Xiao Fu, Vikas Soni, Enusha Karunasena.

**Funding acquisition:** Enusha Karunasena.

**Investigation:** Sang Hyuk Lee, William Kastor, Vikas Soni.

**Methodology:** Sang Hyuk Lee, William Kastor, Xiao Fu, Vikas Soni, Michael Keidar, Marc Donohue, Steve Wood, Enusha Karunasena.

**Project administration:** Steve Wood, Enusha Karunasena.

**Resources:** Michael Keidar, Marc Donohue, Steve Wood, Enusha Karunasena.

**Software:** Xiao Fu.

**Supervision:** Michael Keidar, Marc Donohue, Steve Wood, Enusha Karunasena.

**Validation:** Sang Hyuk Lee, William Kastor, Xiao Fu, Vikas Soni, Enusha Karunasena.

**Visualization:** Sang Hyuk Lee, Xiao Fu.

**Writing – original draft:** Sang Hyuk Lee, William Kastor, Xiao Fu, Vikas Soni.

**Writing – review & editing:** Sang Hyuk Lee, Enusha Karunasena.

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
