## [Decision Letter · Decision Letter 0]

30 May 2024

PONE-D-24-08899Development and Application of Decontamination Methods for the Re-Use of Laboratory Grade Plastic Pipette TipsPLOS ONE

Dear Dr. Lee,

Thank you for submitting your manuscript to PLOS ONE. After careful consideration, we feel that it has merit but does not fully meet PLOS ONE’s publication criteria as it currently stands. Therefore, we invite you to submit a revised version of the manuscript that addresses the points raised during the review process.

**Please address the issues raised by the honorable reviewer. **

We look forward to receiving your revised manuscript.

Kind regards,

Abdul Rauf Shakoori

Academic Editor

PLOS ONE

Journal Requirements:

We would like to thank the ORISE fellowship program for supporting scholars who contributed to this study and FDA Office of the Chief Scientists (OCS)/Medical Counter Measures (MCMi)/OCET and FDA/OST for their funding and support of this study. 

Reviewers' comments:

Reviewer's Responses to Questions

**Comments to the Author**

1. Is the manuscript technically sound, and do the data support the conclusions?

Reviewer #1: Partly

Reviewer #2: Yes

2. Has the statistical analysis been performed appropriately and rigorously? 

Reviewer #1: N/A

Reviewer #2: Yes

3. Have the authors made all data underlying the findings in their manuscript fully available?

Reviewer #1: Yes

Reviewer #2: No

4. Is the manuscript presented in an intelligible fashion and written in standard English?

Reviewer #1: Yes

Reviewer #2: Yes

5. Review Comments to the Author

**Reviewer #1:** General Comment.

The comparison between traditional methods of cleaning, like detergents, and more recent ones, like UV or CAP, is a subject of outmost interest and value in public health. For instance, CAP have been extensively researched in the last 20 years for different applications, but its actual implementation has evolved more slowly, precisely due to the requirement of more studies about its performance.

The paper has strong points like the organized structure of the manuscript, the thorough description of the protocol and elements used in the biological experiments, as well as the presentation of the results (graphs and tables).

My main concern with the manuscript is that I do not think the authors did a good job in comparing the performance of the three methods. There are parameters that are not consistent, like the number of samples, or seem to be chosen arbitrarily. In addition, there is the perception of lack of knowledge in the explanation of the results that were obtained. Although this is not a physics paper, the authors should have a minimum basic knowledge of the methods they are comparing.

Finally, the paper would benefit from a review by a native speaker who could improve the text and make it more natural and fluid.

Specific comments.

Line 45 and 48: “the highest turnover ratio (95.9 %) ... the second highest turnover ratio (98.4 %)”. There is an incoherence in that statement.

Line 48: 14400 PPM * minutes, remove the ‘s’.

Lines 50 to 52: “Relatively, lower turnover ratio and log reduction of CAP could be attributed to development/optimization of treatment conditions, including increases in exposure time and relative to tip positioning.” It is very difficult to make sense of this sentence, please rewrite.

Line 73: instead of using “[8]; Alconox® is a widely used laboratory cleaner” I would suggest using “[8], which is a widely used laboratory cleaner.” to make the reading more fluid.

Lines 75 and 76: The authors wrote “ozone is a reactive allotrope of oxygen with a lone unpaired electron (free radical) that dimerize with valence electrons on other molecules.” That statement is not correct, please check.

Lines 84 and 85: The authors describe CAP as “a thermal non-equilibrium state between heavy positive ions and electrons achieved [10, 11] through rapid atmospheric pressure discharge.” The statement is not entirely accurate. The presence of negative ions in plasma is very important for Electronegative gases like oxygen and therefore, air. In the method presented in the manuscript, plasma is ignited first using helium gas. However, the decontamination effect is achieved through reactive species of oxygen and nitrogen formed through ionization of the surrounding air.

In addition, they should explain briefly and in a very simple (basic) form how the CAP machine works, or its principle of operation (DBD). In lines 243 and 244 they mention the voltage and frequency used, but without said explanation, that information is entirely useless for the reader.

Line 87: “CAP maintains temperatures of 25- 45°C…”. Please, add references.

Lines 94 to 96: The paragraph should be rewritten, because it is not easy to understand.

Lines 245 to 247: The authors mentioned that the pipette tips were placed 2 to 5 cm under the plasma jet for 1 minute. How and why such distance and exposure time were selected.

Line 368 (table 4): Why are there inconsistencies in the number of samples used (73 vs 63)? The turnover ratio is better for O3 14400 ppm than for detergent, but the authors claim otherwise.

I do not think the authors are making a fair comparison or assessment of the cleaning methods proposed, since they seem to have chosen arbitrary exposure distance and time for CAP. They soaked the tip racks for 30 minutes in the detergent solution and used the same time exposure for O3 14400 ppm; however, for CAP they exposed the pipettes for 1 minute, without further explanation to justify that decision. Furthermore, as mentioned before, CAP decontamination is mostly achieved through the action of ROS/RNS (O3 being among them). UV radiation may have some effect if the object is placed near enough to the plasma luminescence. It would have been very useful if the authors could have roughly measured the concentration of at least some of the ROS/RNS, although I understand that this is only possible if they have access to some specialized equipment. However, the authors at least should have done a preliminary study to establish a minimum exposure time, after which the variations in log reduction are not considerable.

**Reviewer #2: **The experiment and data presented is good to be accepted as it is. The experiment and statistical analysis were done. A clear disinfection protocol was also outlined. Only the figures need to be in high quality for publication. Others are okay.

6. PLOS authors have the option to publish the peer review history of their article (what does this mean?). If published, this will include your full peer review and any attached files.

Reviewer #1: No

Reviewer #2: **Yes: **Mohd Ridha Muhamad

---

## [Author Response · Author response to Decision Letter 0]

8 Jul 2024

Hello. Editors of PLOS ONE, 

We appreciate the recommendations to improve our work. 

Please let me know if there is anything we can change.

Thank you for your care and consideration for this manuscript.

Sincerely, 

Sang Hyuk Lee PhD

---

## [Editor Report · Decision Letter 1]

12 Jul 2024

Development and Application of Decontamination Methods for the Re-Use of Laboratory Grade Plastic Pipette Tips

PONE-D-24-08899R1

Dear Dr. Lee,

We’re pleased to inform you that your manuscript has been judged scientifically suitable for publication and will be formally accepted for publication once it meets all outstanding technical requirements.

Kind regards,

Abdul Rauf Shakoori

Academic Editor

PLOS ONE
---

## [Editor Report · Acceptance letter]

9 Dec 2024

PONE-D-24-08899R1 

PLOS ONE

Dear Dr. Lee, 

I'm pleased to inform you that your manuscript has been deemed suitable for publication in PLOS ONE. Congratulations! Your manuscript is now being handed over to our production team.

Kind regards, 

on behalf of

Prof. Dr. Abdul Rauf Shakoori 

Academic Editor

PLOS ONE